# Structural Pruning of Pre-trained Language Models via Neural Architecture Search

**Aaron Klein**                                                       *kleiaaro@amazon.com*
*Amazon Web Services*

**Jacek Golebiowski**                                                 *jacekgo@amazon.com*
*Amazon Web Services*

**Xingchen Ma**                                                       *xgchenma@amazon.com*
*Amazon Web Services*

**Valerio Perrone**                                                   *vperrone@amazon.com*
*Amazon Web Services*

**Cedric Archambeau**                                    *cedric.archambeau@helsing.ai*
*Helsing*

**Reviewed on OpenReview:** *https://openreview.net/forum?id=XiK8tHDQNX*

## Abstract

Pre-trained language models (PLM), for example BERT or RoBERTa, mark the state-of-the-art for natural language understanding task when fine-tuned on labeled data. However, their large size poses challenges in deploying them for inference in real-world applications, due to significant GPU memory requirements and high inference latency. This paper explores neural architecture search (NAS) for structural pruning to find sub-parts of the fine-tuned network that optimally trade-off efficiency, for example in terms of model size or latency, and generalization performance. We also show how we can utilize more recently developed two-stage weight-sharing NAS approaches in this setting to accelerate the search process. Unlike traditional pruning methods with fixed thresholds, we propose to adopt a multi-objective approach that identifies the Pareto optimal set of sub-networks, allowing for a more flexible and automated compression process.

## 1 Introduction

Pre-trained language models (PLMs) such as BERT (Devlin et al., 2019) or RoBERTa (Liu et al., 2019b) are widely used for natural language understanding (NLU) tasks when large amount of labelled data is available for fine-tuning. However, deploying PLMs for inference can be challenging due to their large parameter count. They demand significant GPU memory and exhibit high inference latency, making them impractical for many real-world applications, for example when used in an end-point for a web service or deployed on an embedded systems. Recent work (Blalock et al., 2020; Kwon et al., 2022; Michel et al., 2019; Sajjad et al., 2022) demonstrated that in many cases only a subset of the pre-trained model's parameters significantly contributes to the downstream task performance. This allows for compressing the model by pruning parts of the network while minimizing performance deterioration.

Unstructured pruning (Blalock et al., 2020) computes a score for each weight in the network, such as the weight's magnitude, and removes weights with scores below a predetermined threshold. This approach often achieves high pruning rates with minimal performance degradation, but it also leads to sparse weight matrices, which are not well-supported by commonly used machine learning frameworks. Structured pruning (Michel et al., 2019; Sajjad et al., 2022) removes larger components of the network, such as layers or

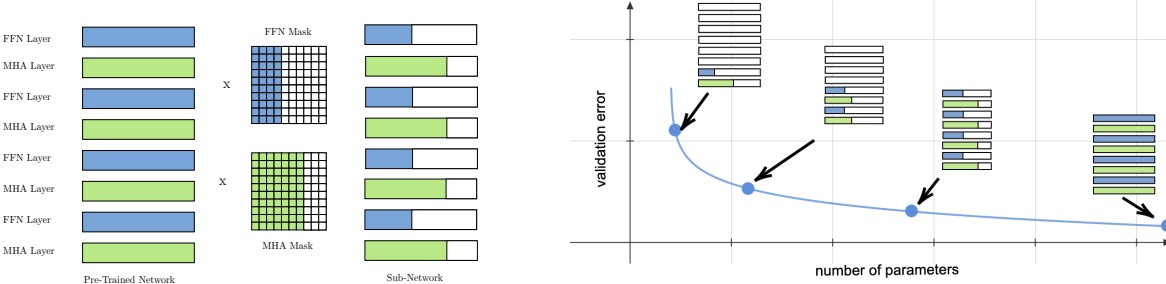

(a) Extracting sub-networks from pre-trained network.  (b) Pareto front of sub-networks.

Figure 1: Illustration of our approach. a) We fine-tune the pre-trained architecture by updating only sub-networks, which we select by placing a binary mask over heads and units in each MHA and FFN layer. b) Afterwards, we run a multi-objective search to select the optimal set of sub-networks that balance parameter count and validation error.

heads. Although it typically does not achieve the same pruning rates as unstructured pruning, it only prunes entire columns/rows of the weight matrix, making it compatible with popular deep learning frameworks and hardware.

Neural architecture search (Zoph & Le, 2017; Real et al., 2017; Bergstra et al., 2013) (NAS) finds more resource efficient neural network architectures in a data-driven way by casting it as an optimization problem. To reduce the computational burden of vanilla NAS, which needs to train and validate multiple architectures, weight-sharing-based neural architecture search (Pham et al., 2018; Liu et al., 2019b; Elsken et al., 2018) first trains a single large network, called the *super-network*, and then searches for *sub-networks* within the super-network.

We propose to use NAS for structural pruning of pre-trained networks, to find sub-networks that sustain performance of the pre-trained network after fine-tuning (see Figure 1 for an illustration). Most structural pruning approaches prune the networks based on a predefined threshold on the pruning ratio. In scenarios where there is no strict constraint on model size, it can be challenging to define such a fixed threshold in advance. NAS offers a distinct advantage over other pruning strategies by enabling a *multi-objective approach* to identify the Pareto optimal set of sub-networks, which captures the *nonlinear relationship* between model size and performance instead of just obtaining a single solution. This allows us to automate the compression process and to select the best model that meets our requirements post-hoc after observing the non-linear Pareto front, instead of running the pruning process multiple rounds to find the right threshold parameter.

While there is a considerable literature on improving the efficiency of PLMs, to the best of our knowledge there is no work yet that explored the potential of NAS for *pruning* fine-tuned PLMs. Our contributions are the following:

- We discuss the intricate relationship between NAS and structural pruning and present a NAS approach that compresses PLMs for inference after fine-tuning on downstream tasks, while minimizing performance deterioration. Our focus lies *not* in proposing a novel NAS method per se, but rather in offering a *practical use-case* for NAS in the context of PLM that works competitively to structural pruning methods from the literature.

- We propose four different search spaces with varying complexity to prune components of transformer based PLM and discuss how their definition affect the structure of sub-networks. Two of these search spaces are typically used by existing structural pruning approaches (see Section 4.2). While one of these commonly used search spaces exhibits the highest degree of freedom, we show in Section 4.1.1 that a search space with lower complexity can be more efficient to explore and eventually lead to better performance within a reasonable budget.

- We contribute a benchmarking suite for multi-objective NAS. We also show how we can apply recently proposed weight-sharing based NAS methods in this setting. Based on our benchmarking suite, we perform a thorough ablation study of standard and weight-sharing based NAS. In the long run we anticipate that our work will drive the development of future NAS methods.

We present an overview of related work in Section 2 and describe our methodology in Section 3. Section 4 provides an empirical comparison of our proposed approach with other structural pruning methods from the literature, along with an in-depth ablation study. Code is available at https://github.com/whittle-org/plm_pruning.

## 2 Related Work

**Neural Architecture Search** (NAS) (see Elsken et al. (2018) for an overview) automates the design of neural network architectures to maximize generalization performance and efficiency (e.g., in terms of latency, model size or memory consumption). The limiting factor of conventional NAS is the computational burden of the search, which requires multiple rounds of training and validating neural network architectures (Zoph & Le, 2017; Real et al., 2017). To mitigate this cost, various approaches have been proposed to accelerate the search process. For example, some of these methods early terminate the training process for poorly performing configurations (Li et al., 2018) or extrapolating learning curves (White et al., 2021b). Weight-sharing NAS (Pham et al., 2018; Liu et al., 2019a) addresses the cost issue by training a single super-network consisting of all architectures in the search space, such that each path represent a unique architecture. Initially, Liu et al. (2019a) framed this as a bi-level optimization problem, where the inner objective involves the optimization of the network weights, and the outer objective the selection of the architecture. After training the super-network, the best architecture is selected based on the shared weights and then re-trained from scratch. However, several papers (Li & Talwalkar, 2020; Yang et al., 2020) reported that this formulation heavily relies on the search space and does not yield better results than just randomly sampling architectures. To address this limitation, Yu et al. (2020) proposed a two-stage NAS process. In the first stage, the super-network is trained by updating individual sub-networks in each iteration, instead of updating the entire super-network. After training, the final model is selected by performing gradient-free optimization based on the shared weights of the super-network, without any further training. Concurrently, Cai et al. (2020) applies a similar approach for convolutional neural networks in the multi-objective setting by first training a single super-network and then searching for sub-networks to minimize latency on some target devices. Related to our work is also the work by Xu et al. (2021), which searches for more efficient BERT architectures during the pre-training phase.

**Structural Pruning** involves removing parts of a trained neural network, such as heads (Michel et al., 2019), or entire layers (Sajjad et al., 2022), to reduce the overall number of parameters while preserving performance. Individual components are pruned based on a specific scoring function, using a manually defined threshold. For transformer-based architectures, Michel et al. (2019) observed that a significant number of heads, up to a single head in a multi-head attention layer, can be deleted after fine-tuning without causing a significant loss in performance. Voita et al. (2019) proposed L0 regularization as a means to prune individual heads in a multi-head attention layer. Kwon et al. (2022) prunes individual heads and units in the fully-connected layers after fine-tuning according to the Fisher information matrix. Sajjad et al. (2022) demonstrated that it is even possible to remove entire layers of a pre-trained network prior to fine-tuning, with minimal impact on performance. In comparison to our data-driven approach, Sajjad et al. (2022) suggested using predefined heuristics (e.g., deleting top / odd / even layers) to determine layers to prune. However, as shown in our experiments, the appropriate architecture depends on the specific task, and more data-driven methods are necessary to accurately identify the best layers to prune.

**Distillation** (Hinton et al., 2015) trains a smaller student model to mimic the predictions of a pre-trained teacher model. For instance, Sanh et al. (2020) used this approach to distill a pre-trained BERT model (Devlin et al., 2019) into a smaller model for fine-tuning. Jiao et al. (2019) proposed a knowledge distillation approach specifically for transformer-based models, which first distills from a pre-trained teacher into a smaller model and then performs task-specific distillation in a second step based on a task augmented dataset. Related to our method is also AdaBERT (Chen et al., 2020) which trains task-specific convolu-

tional neural networks based on differentiable NAS (Liu et al., 2019a) by distilling the knowledge of a PML such as BERT.

Unlike pruning-based methods, distillation allows for complete architectural changes beyond merely dropping individual components. However, from a practical standpoint, determining the optimal structure and capacity of the student network needed to match the performance of the teacher network also amounts to a hyperparameter and neural architecture search problem. Additionally, training a student network requires a significant amount of computational resources. For example, the model by Sanh et al. (2020) was trained for around 90 hours on 8 16GB V100 GPUs. This cost can be amortized by fine-tuning the student model to solve many different tasks, but, depending on the downstream tasks, it potentially requires a substantial amount of iterations which is not always desirable for practitioners who aim to solve a single specific task. This is especially important in the multi-objective setting where many networks need to be distilled to cover the full size/accuracy Pareto front.

**Quantization** (Dettmers et al., 2022; Dettmers & Zettlemoyer, 2023) reduces the precision of model parameters from floating-point numbers to lower bit representations (e.g., 8-bit integers). The main advantage of quantization is the reduction in memory footprint. However, as we show in the Appendix F, this does not necessarily lead to faster latency. Quantization is independent of our NAS approach and can be employed on the pruned network to further decrease memory usage.

## 3 Structural Pruning via Neural Architecture Search

We first provide a multi-objective problem definition for structural pruning of fine-tuned PLMs via neural architecture search. Afterwards, we describe how we can apply weight-sharing based NAS. At the end, we present four search spaces to prune transformer-based architectures, which exhibit a different degree of pruning.

### 3.1 Multi-Objective Sub-Network Selection

We consider a pre-trained transformer model based on an encoder-only architecture, such as for example BERT (Vaswani et al., 2017), with $L$ non-embedding layers, each composed of a multi-head attention (MHA) layer followed by a fully connected feed forward (FFN) layer. However, all methods presented here can also be applied to decoder or encoder-decoder based architectures. Given an input sequence $\boldsymbol{X} \in \mathbb{R}^{n \times d_{model}}$, where $n$ represents the sequence length and $d_{model}$ the size of the token embedding, the MHA layer is defined by: $MHA(\boldsymbol{X}) = \sum_i^H Att(\boldsymbol{W}_Q^{(i)}, \boldsymbol{W}_K^{(i)}, \boldsymbol{W}_V^{(i)}, \boldsymbol{W}_O^{(i)}, \boldsymbol{X})$ where $\boldsymbol{W}_Q^{(i)}, \boldsymbol{W}_K^{(i)}, \boldsymbol{W}_V^{(i)} \in \mathbb{R}^{d_{model} \times d}$ and $\boldsymbol{W}_O^{(i)} \in \mathbb{R}^{Hd \times d_{model}}$ are weight matrices. $Att(\cdot)$ is a dot product attention head (Bahdanau et al., 2015) and $H$ is the number of heads. The output is then computed by $\boldsymbol{X}_{MHA} = LN(\boldsymbol{X} + MHA(\boldsymbol{X}))$, where LN denotes layer normalization (Ba et al., 2016). The FFN layer is defined by $FFN(\boldsymbol{X}) = \boldsymbol{W}_1 \sigma(\boldsymbol{W}_0 \boldsymbol{X})$, with $\boldsymbol{W}_0 \in \mathbb{R}^{U \times d_{model}}$ and $\boldsymbol{W}_1 \in \mathbb{R}^{d_{model} \times U}$, where $U$ denotes the intermediate size and $\sigma(\cdot)$ is a non-linear activation function. Also here we use a residual connection to compute the final output: $x_{FFN} = LN(\boldsymbol{X}_{MHA} + FFN(\boldsymbol{X}_{MHA}))$.

We define a binary mask $\boldsymbol{M}_{head} \in \{0, 1\}^{L \times H}$ for each head in the multi-head attention layer and a binary mask $\boldsymbol{M}_{neuron} \in \{0, 1\}^{L \times U}$ for each neuron in the fully-connected layers. The output of the $l$-th MHA layer and FFN layer is computed by $MHA_l(\boldsymbol{X}) = \sum_i^H \boldsymbol{M}_{head}[i, l] Att(\cdot)$ and $FFN_l(\boldsymbol{X}) = W_1 \circ \boldsymbol{M}_{neuron}[l, :]\sigma(W_0 \boldsymbol{X})$, respectively, where $\circ$ denotes element-wise multiplication.

Now, let's define a search space $\boldsymbol{\theta} \in \Theta$ that contains a finite set of configurations to define possible sub-networks sliced from the pre-trained network. We define a function CREATEMASK that maps from a configuration $\boldsymbol{\theta} \rightarrow \boldsymbol{M}_{head}, \boldsymbol{M}_{neuron}$ to binary masks. Let's denote the function $f_0 : \Theta \rightarrow \mathbb{R}$ as the validation error of the sub-network defined by configuration $\boldsymbol{\theta}$ after fine-tuning on some downstream task. To compute the validation score induced by $\boldsymbol{\theta}$ we place corresponding masks $\boldsymbol{M}_{head}, \boldsymbol{M}_{neuron}$ over the network. Additionally, we define the total number of trainable parameter $f_1 : \Theta \rightarrow \mathbb{N}$ of the sub-network. Our goal is to solve the following multi-objective optimisation problem:

$$min_{\boldsymbol{\theta} \in \Theta}(f_0(\boldsymbol{\theta}), f_1(\boldsymbol{\theta})). \tag{1}$$

In the multi-objective setting, there is no single $\boldsymbol{\theta}_\star \in \Theta$ that simultaneously optimizes all $M$ objectives. Let's define $\boldsymbol{\theta} \succ \boldsymbol{\theta}'$ iff $f_i(\boldsymbol{\theta}) \leq f_i(\boldsymbol{\theta}'), \forall i \in [M]$ and $\exists i \in [k] : f_i(\boldsymbol{\theta}) < f_i(\boldsymbol{\theta}')$. We aim to find the *Pareto Set*: $P_f = \{\boldsymbol{\theta} \in \Theta | \nexists \boldsymbol{\theta}' \in \Theta : \boldsymbol{\theta}' \succ \boldsymbol{\theta}\}$ of points that dominate all other points in the search space in at least one objective.

To solve this optimization problem, we can utilize standard multi-objective search methods that are commonly used for NAS, such as random search. Here each function evaluation consists of fine-tuning a sub-network $\boldsymbol{\theta}$ initialized with the pre-trained weights instead of random weights. We can also directly adopt more advanced strategies, such as multi-fidelity NAS, for example MO-ASHA (Schmucker et al., 2021) to accelerate the search process.

## 3.2 Weight-sharing based Neural Architecture Search

Following previous work (Yu et al., 2020; Wang et al., 2021), our weight-sharing based NAS approach consists of two stages: the first stage is to treat the pre-trained model as super-network and fine-tune it on the downstream task. We explore different super-network training strategies from the literature that update only parts of the network in each step, to avoid co-adaption of sub-networks. The second stage, utilizes multi-objective search strategies to approximate the Pareto-optimal set of sub-networks.

### 3.2.1 Super-Network Training

In the standard NAS setting, we would evaluate $f_0(\boldsymbol{\theta})$ by first fine-tuning the sub-network defined by $\boldsymbol{\theta}$ on the training data before evaluating on the validation data. The weights of the sub-network are initialized based on the pre-trained weights. While more recent NAS approaches (Li & Talwalkar, 2020; Klein et al., 2020) accelerate the search process by early stopping poorly performing sub-networks, this still amounts to an optimization process that requires the compute of multiple independent fine-tuning runs.

The idea of two-stage weight-sharing-based NAS (Yu et al., 2020) is to train a single-set of shared weights, dubbed super-network, that contains all possible networks in the search space. After training the super-networks, evaluating $f_0(\boldsymbol{\theta})$ only requires a single pass over the validation data.

We consider the pre-trained network as super-network with shared weights that contains all possible sub-networks $\boldsymbol{\theta} \in \Theta$. To avoid that sub-networks co-adapt and still work outside the super-network, previous work (Yu et al., 2020; Wang et al., 2021) suggested to update only a subset of sub-networks in each stochastic gradient descent step, instead of updating all weights jointly. We adapt this strategy and sample sub-networks according to the sandwich rule (Yu et al., 2020; Wang et al., 2021) in each update step, which always updates the smallest, the largest and $k$ random sub-networks. The smallest and largest sub-network correspond to the lower and upper bound of $\Theta$, respectively. For all search spaces $\Theta$ define below, the upper bound is equal to full network architecture, i.e, the super-network and the lower bound removes all layer except the embedding and classification layer.

Additionally, we use in-place knowledge distillation (Yu et al., 2019) which accelerate the training process of sub-networks. Given the logits $\pi_{supernet}(\boldsymbol{x})$ of the super-network, which we obtain for free with the sandwich rule, and the logits of a sub-network $\pi_{\boldsymbol{\theta}}(\boldsymbol{x})$, the loss function to obtain gradients for the sub-networks follows the idea of knowledge distillation:

$$\mathcal{L}_{KD} = \mathcal{L}_{CE} + D_{\mathrm{KL}}\left(\sigma\left(\frac{\pi_{supernet}}{T}\right), \sigma\left(\frac{\pi_{\boldsymbol{\theta}}}{T}\right)\right), \tag{2}$$

where $D_{\mathrm{KL}}(\cdot)$ denotes the Kullback-Leibler divergence between the logits of the super-network and the sub-network, $T$ a temperature parameter, $\sigma(\cdot)$ the softmax function and $\mathcal{L}_{CE}$ is the cross-entropy loss of the training data.

### 3.2.2 Sub-network selection

After training the super-network, we compute the validation error $f_0(\boldsymbol{\theta})$ by applying $\boldsymbol{M}_{head}$ and $\boldsymbol{M}_{neuron}$ to the shared weights and performing a single pass over the validation data. This substantially reduces the computational cost involved in the multi-objective problem stated in Equation 1. To solve this problem,

we essentially can use the same multi-objective approaches as for standard NAS (see Appendix D), except for multi-fidelity approaches such as MO-ASHA. Multi-fidelity methods early stop the training process by evaluating the model at intermediate time steps, called rung levels. Since we do not perform any additional training steps anymore and just evaluate the final sub-network, these rung levels are not defined.

### 3.3 Search Space

The search space $\Theta$ defines sub-networks of the pre-trained network architecture. An expressive $\Theta$ allows for fine-grained pruning but might also become infeasible to explore. We propose the following search spaces that exhibit different levels of complexity. For each search space we provide pseudo code to define the CREATEMASK function in Appendix B.

- **SMALL**: We define the number of heads $\mathcal{H} = [0, H]$, the number of units $\mathcal{U} = [0, U]$ and the total number of layers $\mathcal{L} = [0, L]$, such that $\Theta = \mathcal{H} \times \mathcal{U} \times \mathcal{L}$. Compared to the other search spaces, the dimensionality of this search space remains fixed with different model sizes, and only its upper bound i.e. $(H, U, L)$ increases. For each layer, we always keep the first $h \in \mathcal{H}$ heads and $u \in \mathcal{U}$ units, respectively, to enforce that CREATEMASK is a bijective mapping (see Appendix B).

- **LAYER**: Inspired by Sajjad et al. (2022), we prune individual attention and fully-connected layers instead of single heads and neurons. We define a search space $\Theta = \{0, 1\}^L$ that contains one binary hyperparameter for each layer that determines if the corresponding layer is removed.

- **MEDIUM**: Based on the previous search space, we allow for a flexible number of heads / units per layer. For each layer $l \in [0, L]$, we define $\mathcal{H}_l = [0, H]$ and $\mathcal{U}_l = [0, U]$, such that the final search space is $\Theta = \mathcal{H}_0 \times \mathcal{U}_0 \ldots \mathcal{H}_L \times \mathcal{U}_L$. As for the SMALL search space we also keep the first heads and units in each layer.

- **LARGE**: For each head and neuron in the fully-connected layer we define a single binary $\Theta_i = \{0, 1\}$ which is combined to form the search space $\Theta = \Theta_0 \times \ldots \times \Theta_{L(H+I)}$. This is the most expressive search space, but also grows quickly with the model size. The search space is also commonly used by other structural pruning approaches (Kwon et al., 2022). It might not be very useful in practice, because we cannot easily remove single rows/columns of the weight matrix with most transformer implementations and hence it will not necessarily reduce the inference latency. However, it provides us a reference in terms of predictive performances that can be retained under a certain pruning ratio.

Each search space induces a different pattern for $\boldsymbol{M}_{head}$ and $\boldsymbol{M}_{neuron}$ that we place over the super-network to select sub-networks (see Figure 2 for some examples). To see how this effects the distribution over parameter count and hence the sampling during the super-network training, we sample $N = 500$ configurations $\{\boldsymbol{\theta}_0, ..., \boldsymbol{\theta}_N\}$ uniformly at random and compute the number of trainable parameters $\{f_1(\boldsymbol{\theta}_i), ..., f_1(\boldsymbol{\theta}_N)\}$ for all four search spaces (see Figure 3). The SMALL search space is somewhat biased to smaller networks. The MEDIUM search space, even though more expressive, is highly biased towards mid-size networks, since on average half of the heads / neurons are masked out. For the two binary search spaces LAYER and LARGE, we can achieve a uniform distribution over the number of parameters, by using the following sampling process. We first sample an integer $k \sim U(0, K)$, where $k = L$ for the LAYER search space, and $k = L(H + I)$ for the LARGE search space. Afterwards, we randomly select $k$ entries of the binary vector $\boldsymbol{\theta} \in \Theta$ and set them to 1.

## 4 Experiments

We evaluate different types of NAS for structural pruning on eight text classification tasks, including textual entailment, sentiment analysis and multiple-choice question / answering. We provide a detailed description of each task in Appendix C. All tasks come with a predefined training and evaluation set with labels and a hold-out test set without labels. We split the training set into a training and validation set (70%/30% split) and use the evaluation set as test set. We fine-tune every network, sub-network or super-network, for

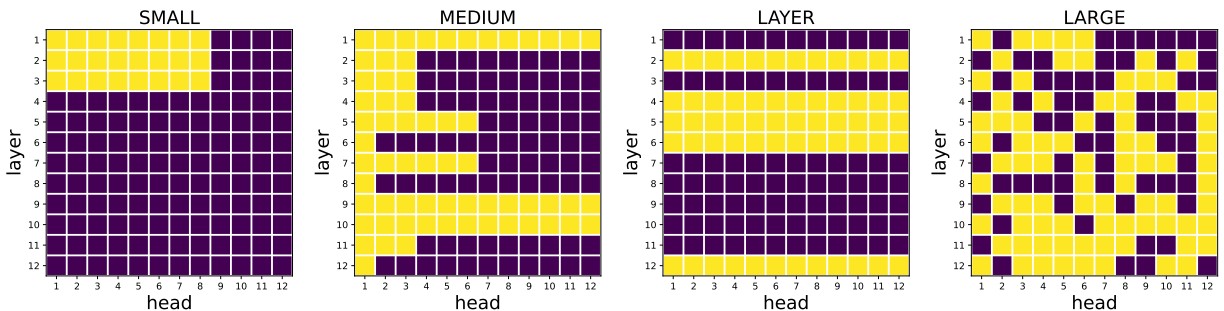

Figure 2: Examples of head masks $\boldsymbol{M}_{head}$ sampled uniformly at random from different search spaces. Dark color indicates that the corresponding head is masked. The same pattern can be observed for $\boldsymbol{M}_{neuron}$

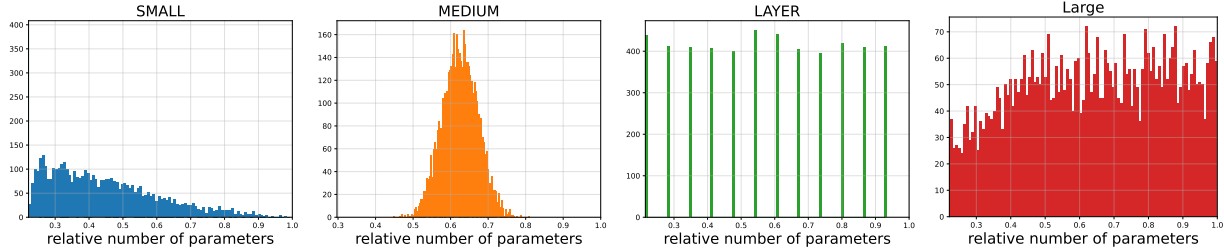

Figure 3: Distribution of the parameter count $f_1(\boldsymbol{\theta})$ for uniformly sampled $\boldsymbol{\theta} \sim \Theta$.

5 epochs on a single GPU. For all multi-objective search methods, we use Syne Tune (Salinas et al., 2022) on a single GPU instance. We use BERT-base (Devlin et al., 2019) (cased) and RoBERTa-base (Liu et al., 2019b) as pre-trained network, which consists of $L = 12$ layers, $I = 3072$ units and $H = 12$ heads (other hyperparameters are described in Appendix A). While arguably rather small for today's standards, they still achieve competitive performance on these benchmarks and allow for a more thorough evaluation. We also present a comparison to quantization in Appendix F.

## 4.1 Benchmarking Neural Architecture Search

We now present an evaluation of different multi-objective NAS approaches on our benchmarking suite. To quantify the performance of a Pareto set $P_f$, we compute for each Pareto set the Hypervolume (Zitzler et al., 2003) and report the regret, i e. the difference to the best possible Hypervolume averaged across all repetitions. Given a reference point $\mathbf{r} \in \mathbb{R}^M$, the Hypervolume $HV(P_f|\mathbf{r}) = \lambda(\cup_{\boldsymbol{y} \in P_f}[\boldsymbol{y}, \mathbf{r}])$ is defined as the $M$-th dimensional Lebesgue measure $\lambda$ between the Pareto set $P_f$ and the reference point $\mathbf{r}$, where $[\boldsymbol{y}, \mathbf{r}]$ represents the hyper-rectangle between $\boldsymbol{y}$ and $\mathbf{r}$ (see Figure 4 for an example).

To compute the Hypervolume, we first normalize each objective based on all observed values across all methods and repetitions via Quantile normalization. This results in a uniform distribution between $[0, 1]$, and we use $r = (2, 2)$ as reference point, which means the highest possible Hypervolume would be 4. We evaluate each method with 10 different seeds for the random number generation.

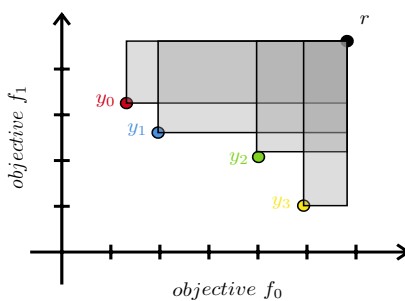

Figure 4: Example to compute the Hypervolume $HV(P_f|\mathbf{r})$, corresponding to the sum of the rectangles, across a reference point $\mathbf{r}$ and a set of points $P_f = \{\mathbf{y_0}, \mathbf{y_1}, \mathbf{y_2}, \mathbf{y_3}\}$

### 4.1.1 Search Space

First, we compare the search spaces definitions from Section 3.3 using weight-sharing based NAS. We fine-tune the super-network as described in Section 3.2 and sample 100 sub-networks uniformly at random to compute the hypervolume.

**Conclusions:** Within this budget (see Figure 5), *the SMALL search space achieves the best performance across all datasets*, except for COLA. Interestingly, even though the *MEDIUM* search space allows for a more fine-grained per layer pruning, it leads to worse results. We attribute this to the non-uniform distribution of parameter count as described in Section 3.3. The *LAYER* search space often out-performs the *MEDIUM* and *LARGE* search space, but, except for COLA, leads to Pareto sets that under-perform compared to the *SMALL* search space. The *LARGE* search space, which is a superset of the other search spaces, *seems infeasible to explore with random sampling* over so few observations. We use the SMALL search space for the remaining experiments.

### 4.1.2 Standard Neural Architecture Search

We compare the following multi-objective search methods to tackle the NAS problem described in Section 3 where each sub-network is fine-tuned in isolation. We provide a more detailed description of each method in Appendix D. A simple *multi-objective local search* (LS) described in Appendix D. This is inspired by the work of White et al. (2021a), which showed that local search often performs competitively on NAS problems. *Random search* (RS) (Bergstra & Bengio, 2012) samples architectures uniformly at random from the search space. A multi-objective version of the *regularized evolution* (REA) algorithm (Real et al., 2019), frequently used in the NAS literature. Compared to the original singe-objective algorithm, we sort elements in the population via non-dominated sorting. *Expected Hypervolume Improvement* (EHVI) (Daulton et al., 2020) is a multi-objective Bayesian optimization strategy that samples candidate points using a Gaussian process model of the objective function. Lastly, we include MO-ASHA (Schmucker et al., 2021), a multi-objective version of *asynchronous successive halving* (Li & Talwalkar, 2020; Jamieson & Talwalkar, 2016) that terminates the training process of poorly performing candidates early to accelerate the overall optimization process. While MO-ASHA could potentially be combined with a model-based approach, as commonly done for single-objective optimization (Falkner et al., 2018; Klein et al., 2020), here we followed the original algorithm and sample candidate uniformly at random from the search space.

Following common experimental practice from the HPO literature, we aggregate results by computing the average ranks of each methods across repetitions, datasets and time steps. Following Feurer et al. (2015), we sample 1000 bootstrap samples across all repetitions and tasks, to compute the rank of each method and average across all samples. Results are shown in Figure 6a. We shows results for each individual task in Appendix E.

**Conclusions:** Somewhat surprisingly *RS is a strong baseline on these benchmarks*, outperforming more sophisticated approaches such as EHVI or MO-REA. Fine-tuning these models is often unstable (Mosbach et al., 2021) especially on smaller datasets, resulting in high observation noise. For the RoBERTa-base model, *LS often performs competitively to RS* given a sufficient large budget. MO-ASHA quickly stops the evaluation of poorly performing sub-networks and hence outperforms RS on average. However, *on small datasets such as RTE, fine-tuning is faster than the non-dominated sorting of MO-ASHA*, such that it converges slower than RS (see Appendix E).

### 4.1.3 Weight-sharing based Neural Architecture Search

Next, we evaluate different techniques for two-stage NAS for this problem. We distinguish between, fine-tuning the super-network and multi-objective search.

**Super-network fine-tuning:** First, we compare the following strategies to fine-tune the super-network from the literature. To compare these methods, after fine-tuning the super-network, we sample 100 sub-networks uniformly at random from the SMALL search space to estimate the Pareto set and report here the

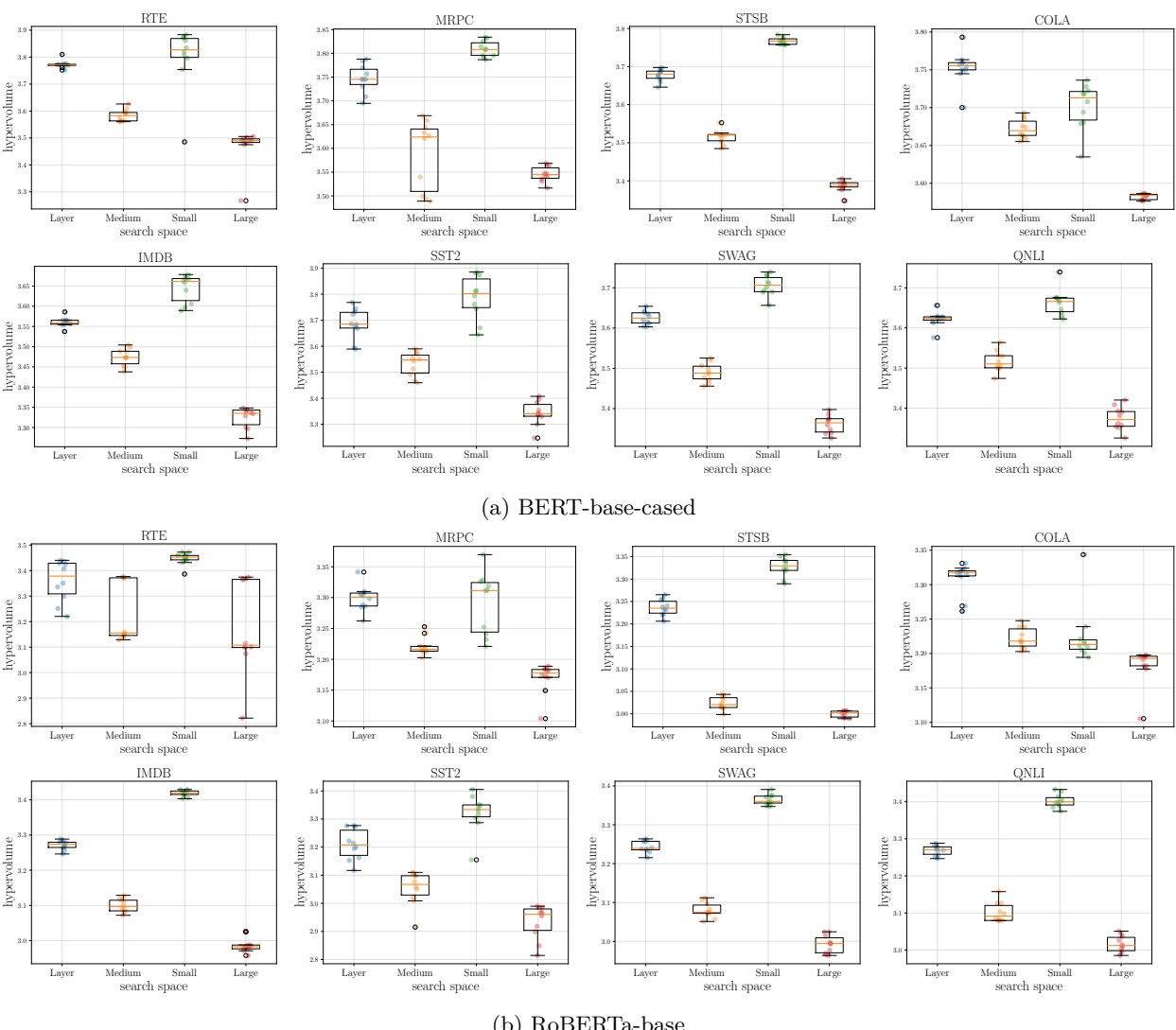

(a) BERT-base-cased

(b) RoBERTa-base

Figure 5: Comparison of the four different search spaces using weight-sharing based NAS. We sample 100 random sub-networks uniformly at random using the fine-tuned weights of the super-network. The SMALL search space dominates the other search spaces except for the COLA dataset. While SMALL is a subset of MEDIUM and LARGE, these spaces are too high-dimensional to be explored with a sensible compute budget. First two rows show results for BERT-base-cased and last two rows for RoBERTa-base.

Hypervolume. We repeat this process 10 times with a different seed for the random number generation. For each repetition we use the exact same set of random sub-networks for all super-network training strategies.

- **standard**: Which trains all weights of super-network in the standard fine-tuning setting

- **random**: Samples a single random sub-network in each update steps

- **random-linear**: Inspired by Bender et al. (2018), we either sample a random sub-network with probability $p$ or the full-network with probability of $1 - p$ in each update step. Thereby, $p$ is linearly increased from 0 to 1 after each update step over the course of training.

- **sandwich**: The super-network is updated according to the sandwich rule (Yu et al., 2020; Wang et al., 2021) described in Section 3.2. We set the number of random sub-networks in each update step to $k = 2$.

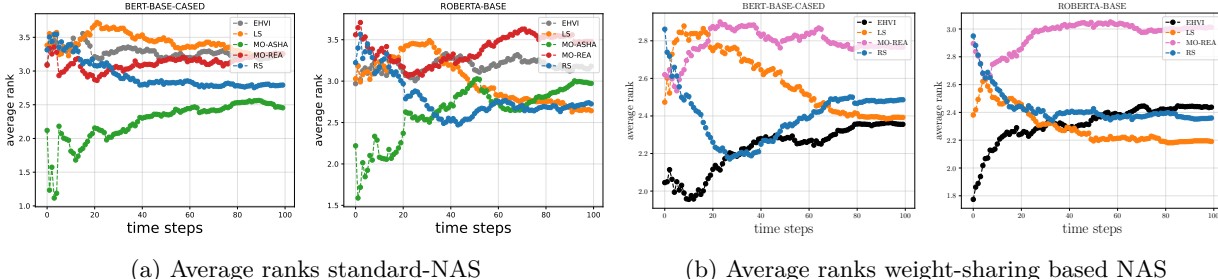

(a) Average ranks standard-NAS      (b) Average ranks weight-sharing based NAS

Figure 6: Average ranks of the different multi-objective search methods across all dataset for a) standard-NAS and b) weights-sharing based NAS for both BERT-based cased and RoBERTa-base models. Random search based approaches (RS, MO-ASHA)perform competitively in the standard NAS setting. EHVI outperforms other methods at earlier time steps which we attribute to its probabilistic model. Given a sufficient amount of time often finds well-performing Pareto fronts.often finds a well-perofforming Pareto fronts.

- **kd**: Update $k = 2$ random sub-networks using in-place knowledge distillation (Yu et al., 2020; Wang et al., 2021) according to Equation 2.

- **full**: Implements the training protocol described in Section 3.2, i.e it combines the sandwich rule with in-place knowledge distillation to update sub-networks.

**Conclusions:** Figure 7 shows the Hypervolume across all task for BERT-base and RoBERTa-base, respectively. *Standard fine-tuning and just randomly sampling a sub-network leads to poorly performing Pareto sets* compared to other methods. The only exception is the COLA dataset, where standard fine-tuning sometimes works best. However, we also observe high variation across runs on this datasets. Linearly increasing the probability of sampling a random sub-networks improves to just random sampling. *Better results are achieved by using the sandwich rule or knowledge distillation.* Thereby, combining both slightly improves results further.

**Multi-Objective Search** Lastly, we compare in Figure 6b average ranks of the same multi-objective search methods as for standard-NAS. We follow the same process as described in Section 4.1.2 to compute averange ranks. We do not include MO-ASHA in this setting, since each function evaluation only consists of validating a sub-network based on the shared weights of the super-network after fine-tuning and hence does not allow for a multi-fidelity approach. Optimization trajectories for all datasets are in Appendix E.

**Conclusions:** As for standard NAS, RS is a surprisingly strong baseline. *EHVI performs better at early time-steps.* We found using the shared weights of the super-networks for evaluation results in a much smaller observation noise than fine-tuning each sub-network in isolation, which is less deceiving for the probabilistic model of EHVI. Given enough time, *LS starts outperforming RS and EHVI on RoBERTa-base model* and performs competitively to EHVI on the BERT-base model.

## 4.2 Comparison to other Structural Pruning Approaches

We now present a comparison against other structural pruning approaches. For NAS we use the SMALL search space defined in Section 3.3 based on our ablation study in Section 4.1. Each method had the same total amount of wall-clock time and compute, which include both fine-tuning and search. We compare the following methods:

- **Head-Pruning** (HP) (Michel et al., 2019) prunes heads greedily using a proxy score for the importance of each head for final performance based on the gradient of the loss function.

- **Retraining Free Pruning** (RFP) (Kwon et al., 2022) uses a three-phased pruning strategy that, based on a threshold $\alpha$, prunes individual heads in the MHA layer and units in the FFN layer.

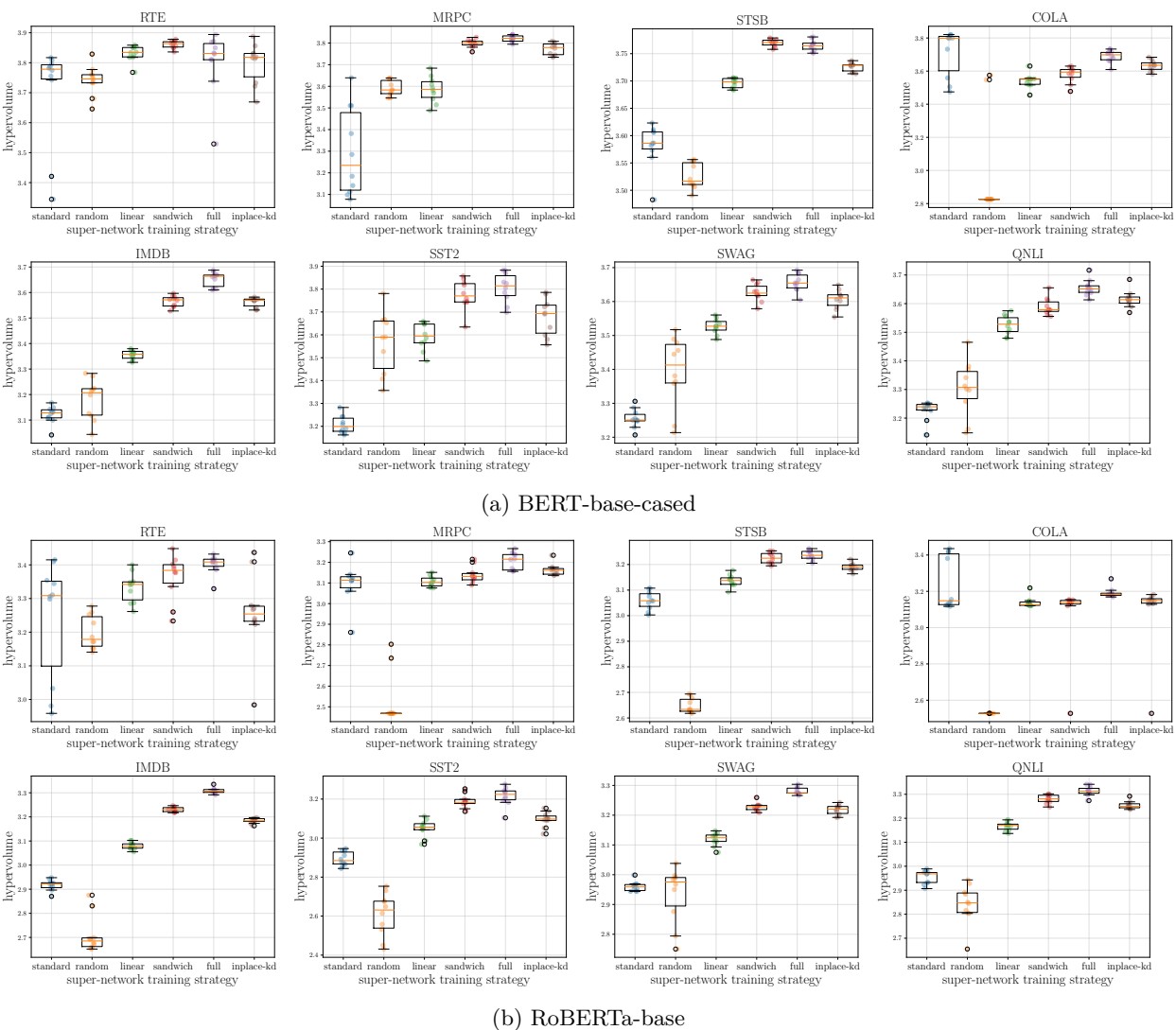

(a) BERT-base-cased

(b) RoBERTa-base

Figure 7: Comparison of different strategies to fine-tune the super-network. First two rows show BERT-base and last two rows show RoBERTa-base.

The first phase computes a binary mask for heads and units by computing the diagonal Fisher information matrix. The matrix is then rearranged by a block-approximated Fisher information matrix. In the last step, the masked is further tuned by minimizing the layer-wise reconstruction error. This method operates in the LARGE search space described in Section 3.3. We run RFP with different values for $\alpha \in \{0.1, 0.2, ..., 0.9\}$ to obtain a Pareto set of architectures.

- **Layer Dropping** (LD): Following Sajjad et al. (2022) we first remove the top $n \in 1, ..., L - 1$ layers and fine-tune the remaining layers directly on the downstream task. To obtain a Pareto set of $N$ points, we fine-tune $N$ models with different amount of layers removed. This method serves as a simple heuristic to explore the LAYER search space.

- **Standard NAS** (S-NAS): As for the previous experiments, we used standard NAS using random search where each sub-network is initialized with the pre-trained weights and then fine-tuned independently.

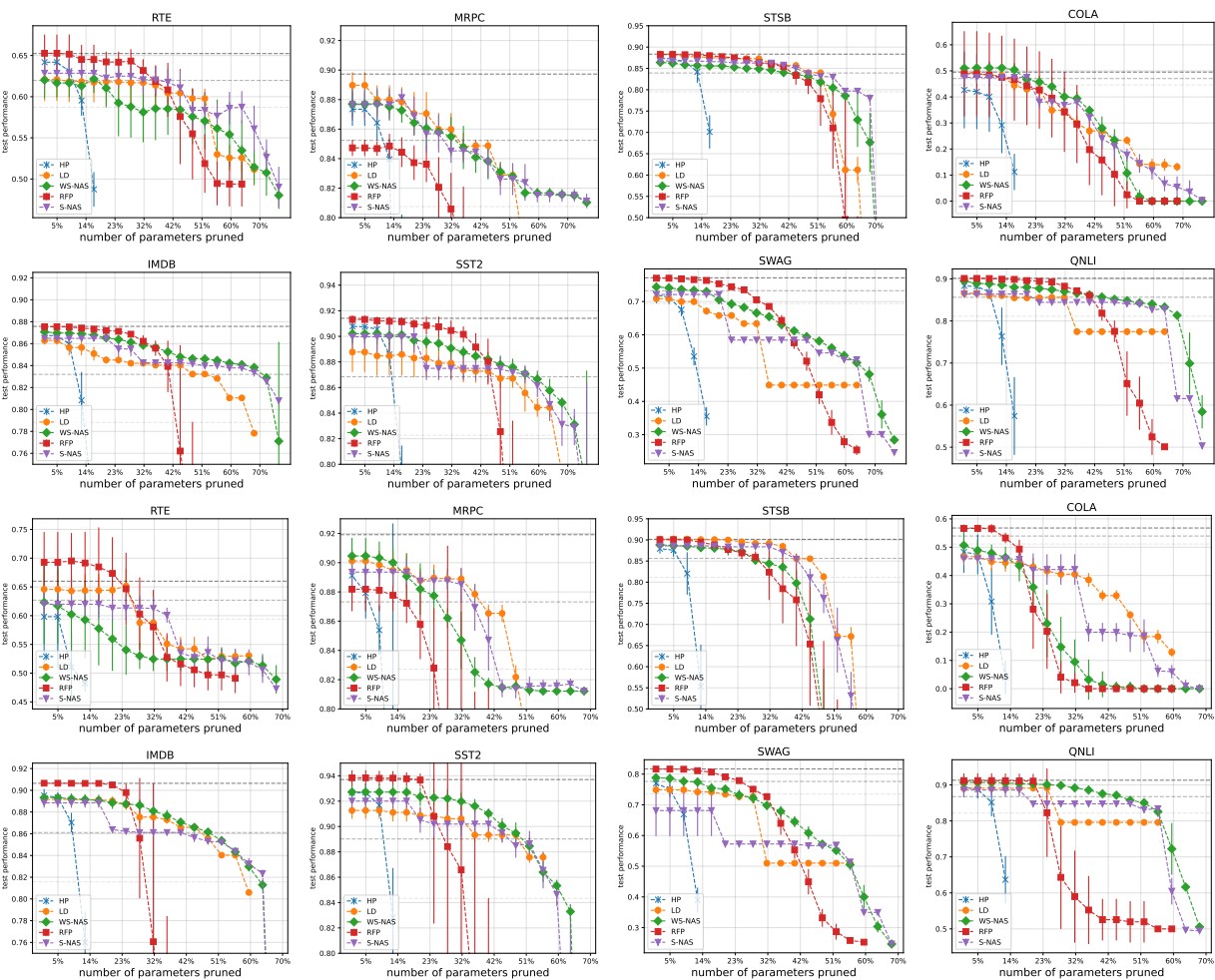

Figure 8: Loss in test performance (the higher the better) versus the parameter count relative to the unpruned model on all 8 text classification datasets. First two rows show BERT-base and last two rows show RoBERTa-base.

- **Weight-sharing NAS** (WS-NAS): Follows the two-stage weight sharing based NAS approach outlined in Section 3.2. We use sandwich rule and in-place knowledge distillation to train the supernetwork and EHVI to search for the Pareto optimal set of sub-networks.

To compare results, we normalize the number of parameters to $[0, 1]$ and bin results based on different thresholds $\beta \in \{0.2, ...0.9\}$. Note that roughly 20% of the parameters of BERT-base / RoBERTa-base are included in the embedding and classification head, and hence cannot be trivially pruned without changing the embedding dimension or the number of classes. For each bin, we report the best performance of the solution with $\leq \beta$ parameters. We discuss the relationship between parameter count and model inference time in F.

Figure 8 shows the parameter count (horizontal axis) and the test error (vertical axis) relative to the unpruned network for all datasets. For reference, we indicate 95% and 90% relative performance to the unpruned network as well as the original performance by dashed lines. We sort dataset by their size from left to right.

**Conclusion:** First, both *S-NAS and WS-NAS achieve competitive performance to structural pruning methods*. Especially for *higher pruning ratios, both S-NAS and WS-NAS outperform RFP and HP even though they operate in the LARGE search space*, which is much higher dimensional than the SMALL search space used for NAS. This indicates that these methods cannot rigorously handle such high dimensional spaces.

Second, *NAS methods seems to perform better for larger datasets.* Given a sufficient amount of budget, simple LD, which operates in the LAYER search space, also achieves competitive results. This is in-line with our results in Section 4.1.1, showing that the LAYER search space can still provide sensible results compared to more expressive search spaces.

WS-NAS substantially reduces the overall runtime by fine-tuning only a single super-network, which enables it to achieve much better performance with a limited amount of compute compared to S-NAS. However, we expect S-NAS to outperform WS-NAS if we scale the total compute, since each sub-network if s fine-tuned in isolation and hence able to optimally adapt to the training data.

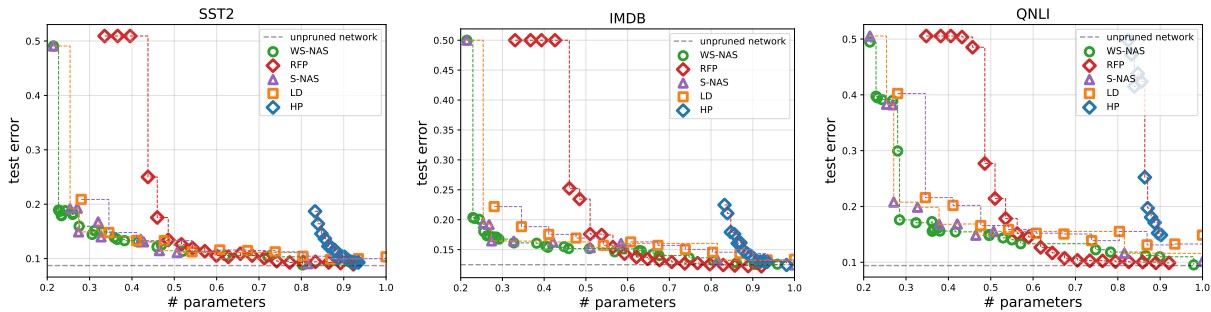

Figure 9: Pareto fronts of single runs for each method for SST2, IMDB and QNLI dataset to balance parameter-count and test-error (the lower the better).

For a qualitative comparison, we show the results for a single run on three datasets in Figure 9 right. More example runs are shown in Appendix E.

## 4.3 Conclusions

We propose NAS for structural pruning of fine-tuned PLMs. By utilising a multi-objective approach, we can find the Pareto optimal set of sub-networks that balance between model size and validation error. Returning a Pareto set of sub-networks allows practitioners to select the optimal network without running the pruning process multiple times with different thresholds. We also provide an in-depth analysis of recently developed two-stage weight-sharing approaches in this setting, which require only a single fine-tuning run of the PLM.

Future work could explore the instruction tuning (Wei et al., 2022) setting, where the final model is evaluated in a few-shot setting. Our approach samples sub-networks uniformly at random, which allocates the same amount of update steps to all sub-networks on average. Future work could explore more complex sampling distribution biased towards sub-networks closer to the Pareto set.

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

## A  Hyperparameters

Table A shows the hyperparameters for fine-tuning the super-network. We largely follow default hyperparameters recommended by the HuggingFace transformers library. For all multi-objective search method, we follow the default hyperparameter of Syne Tune.

| Hyperparameter | Value |
|---|---|
| Learning Rate | 0.00002 |
| Number of random sub-networks $k$ | 2 |
| Temperature $T$ | 10 |
| Batch Size | 4 |

## B  Masking

Algorithm 1, 2, 3 and 4 show pseudo code for the LAYER, SMALL, MEDIUM and LARGE search space, respectively. Note that, $\mathbf{1}$ indicates a vector of ones. For a matrix $\boldsymbol{M}$, we write $\boldsymbol{M}[:, :N]$ to denote the first $N$ columns for all rows and, vice versa, $\boldsymbol{M}[:N, :]$ for the first $N$ rows.

**input** : sub-network configuration $\boldsymbol{\theta} \in \{0, 1\}^L$
**output:** $\boldsymbol{M}_{head}, \boldsymbol{M}_{neuron}$
$\boldsymbol{M}_{head} \leftarrow [0]^{L \times H}$;
$\boldsymbol{M}_{neuron} \leftarrow [0]^{L \times I}$;
**for** $l = 0, \dots, L - 1$ **do**
    $\boldsymbol{M}_{head}[l, :] \leftarrow \boldsymbol{\theta}[l]$;
    $\boldsymbol{M}_{neuron}[l, :] \leftarrow \boldsymbol{\theta}[l]$;
**end**

**Algorithm 1:** CREATEMASK function for LAYER search space

**input** : sub-network configuration $\boldsymbol{\theta} \in \mathcal{H}_0 \times \mathcal{U}_0 \ldots \mathcal{H}_L \times \mathcal{U}_L$
**output:** $\boldsymbol{M}_{head}, \boldsymbol{M}_{neuron}$
$\boldsymbol{M}_{head} \leftarrow [0]^{L \times H}$;
$\boldsymbol{M}_{neuron} \leftarrow [0]^{L \times I}$;
**for** $l = 0, \ldots, L - 1$ **do**
$\quad\quad h = \boldsymbol{\theta}[2 * l]$ ;                              /* number of heads in layer $l$ */
$\quad\quad u = \boldsymbol{\theta}[2 * l + 1]$ ;                         /* number of units in layer $l$ */
$\quad\quad \boldsymbol{M}_{head}[l, :h] \leftarrow \mathbf{1}$;
$\quad\quad \boldsymbol{M}_{neuron}[l, :u] \leftarrow \mathbf{1}$;
**end**

**Algorithm 2:** CREATEMASK function for MEDIUM search space

**input** : sub-network configuration $\boldsymbol{\theta} \in \mathcal{H} \times \mathcal{U} \times \mathcal{L}$
**output:** $\boldsymbol{M}_{head}, \boldsymbol{M}_{neuron}$
$h = \boldsymbol{\theta}[0]$ ;                                       /* number of heads */
$u = \boldsymbol{\theta}[1]$ ;                                      /* number of units */
$l = \boldsymbol{\theta}[2]$ ;                                      /* number of layers */
$\boldsymbol{M}_{head} \leftarrow [0]^{L \times H}$;
$\boldsymbol{M}_{neuron} \leftarrow [0]^{L \times I}$;
$\boldsymbol{M}_{head}[:l, :h] \leftarrow \mathbf{1}$;
$\boldsymbol{M}_{neuron}[:l, :u] \leftarrow \mathbf{1}$;

**Algorithm 3:** CREATEMASK function for SMALL search space

**input** : sub-network configuration $\boldsymbol{\theta} \in \{0, 1\}^{L * (H + U)}$
**output:** $\boldsymbol{M}_{head}, \boldsymbol{M}_{neuron}$
$\boldsymbol{M}_{head} \leftarrow \boldsymbol{\theta}[:, :H]$;
$\boldsymbol{M}_{neuron} \leftarrow \boldsymbol{\theta}[:, H:]$;

**Algorithm 4:** CREATEMASK function for LARGE search space

## C  Datasets

We use the following 10 dataset test classification datasets. All dataset are classification tasks, except for STSB, which is a regression dataset.

- The Recognizing Textual Entailment (RTE) dataset aims to identify the textual entailment of two sentences.

- The Microsoft Research Paraphrase Corpus (MRPC) dataset consists of sentence pairs extracted from online news sources. The task is to predicts if these pairs are semantically equivalent to each other.

- The Semantic Textual Similarity Benchmark (STSB) consists of sentences pairs that are scored between 1 and 5 based on their similarity.

- The Corpus of Linguistics Acceptability (COLA) dataset contains English sentences that are labeled as grammatically correct or not.

- The IMDB dataset for sentiment classification (positive / negative) of movie reviews.

- The Stanford Sentiment Treebank (SST2) datasets classifies the positive / negative sentiment of sentences extracted from movie reviews.

- Situations With Adversarial Generations (SWAG) dataset for multiple-choice question / answering.

- QNLI is a modified version of the Stanford Question Answering Dataset which is a collection of question / answer pairs where question are written by human annotators and answers are extracted from Wikipedia. The task is to predict whether the answers is correct.

# D   Additional Details NAS Search

In this section, we present more details about all multi-objective search strategies that we used for both the standard NAS scenario, where each sub-network is fine-tuned from the pre-trained weights, and the weight-sharing scenario, where sub-networks are evaluated using the shared weights of the super-network.

- **Random Search**: We follow the standard random search approach (Bergstra & Bengio, 2012), where for a fixed number of iterations $T$, we sample in each iteration $t \in \{0, ..., T\}$ a random sub-networks uniformly from the search space $\boldsymbol{\theta}_t \sim \mathcal{U}(\Theta)$.

- **MO-Regularized Evolution**: We adapt the original algorithm proposed by (Real et al., 2019) which maintains a population of architectures and, in each iteration, removes the oldest architecture from the population. To sample a new configuration, we follow Real et al. (2019) and first sample a set of random configurations and mutate the configuration from the set with the lowest rank. Instead of just using the validation performance, we rank each element in the population via non-dominated sorting. We mutate an architecture $\boldsymbol{\theta}$ by first sampling a random dimension of the search space $d \sim \mathcal{U}(0, |\boldsymbol{\theta}|)$ and than sample a new value for this dimension $\boldsymbol{\theta}[d] = \mathcal{U}(\Theta_d)$. This follows the same process as for our multi-objective local search described below.

- **Expected Hypervolume Improvement**: Bayesian optimization optimizes a single function $f : \Theta \rightarrow \mathbb{R}$, where in each iteration $t$ we select the most promising candidate in the input space $\boldsymbol{\theta}_\star \in \arg\max a_{p(f|D_t)}(\boldsymbol{\theta})$ according to some acquisition function $a : \Theta \rightarrow \mathbb{R}$. The idea of this acquisition function is to trade-off exploration and exploitation, based on a probabilistic model of the objective function $p(f|D_t)$, trained on some observed data points $D_t = \{(\boldsymbol{\theta}_0, y_0), ...(\boldsymbol{\theta}_t, y_t)\}$, where $y \sim \mathcal{N}(f, \sigma)$. To adapt Bayesian optimization to the multi-objective setting, we follow Daulton et al. (2020) and use a single Gaussian process $p(f|D)$ for all objective functions $f(\boldsymbol{\theta}) = \{f_0(\boldsymbol{\theta}), ..., f_k(\boldsymbol{\theta})\}$. Here, the acquisition function $a(\boldsymbol{\theta}) = \mathbb{E}_{p(f|D)}[HVI(f(\boldsymbol{\theta})]$ computes the expected improvement of the hypervolume $HVI(\boldsymbol{y}) = HV(P_f \cup \boldsymbol{y}, r) - HV(P_f, r)$ based on the current Pareto front $P_f$ and some reference point $r$.

- **Multi-Objective ASHA**: Given a halving constant $\eta$, a minimum $r_{min}$ and maximum $r_{max}$ number of epochs for fine-tuning, successive halving defines a set of rungs $\mathcal{R} = \{r_{min}^k | k = 0, ..., K\}$ where for simplicity we assume that $\frac{r_{max}}{r_{min}} = \eta^K$. Starting from a set of randomly sampled configurations $C = \{\boldsymbol{\theta}_0, ..., \boldsymbol{\theta}_n\}$, successive halving evaluates all configuration on the first rung level $r_0$ and promotes the top $\eta^{-1}$ configuration for the next rung while discarding the others. This process is iterated until we reach the maximum rung level $r_k = r_{max}$. We follow common practice (Li et al., 2017; Falkner et al., 2018), and run multiple rounds of successive halving until we hit a maximum budget (defined in wall-clock time). Asynchronous successive halving (Li et al., 2018)adapts standard successive halving to distributed asynchronous case, which requires some changes in the decision making routing. To cater for the multi-objective setting, we use multi-objective ASHA (Schmucker et al., 2021) which uses non-dominated sorting instead of just the validation accuracy to rank configurations on each rung-level.

- **Multi-objective Local Search**: Previous work (White et al., 2021a) has demonstrated that simple local search often performs competitively compared to more advanced NAS methods. We propose a straightforward multi-objective local search approach. Starting from the current Pareto front $P_f$, which is initialized by some starting point, we randomly sample an element $\boldsymbol{\theta}_\star \sim P_f$ and then generate a random neighbor point by permuting a single random entry of $\boldsymbol{\theta}_\star$.

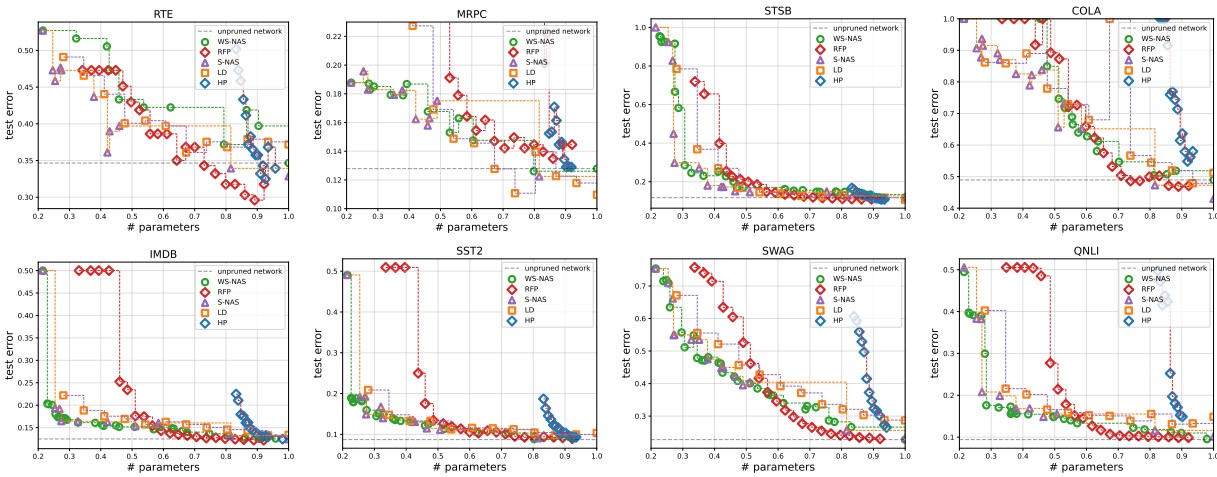

Figure 10: Single Pareto fronts of random run for each method on all 8 GLUE datasets.

**input** : Search space $\Theta$, number of iteration $T$, starting point $\boldsymbol{\theta}_{start}$
**output:** Pareto front $P$
```
/* evaluate starting point                                                    */
```
$P_0 \leftarrow \{\boldsymbol{\theta}_{start}\};$
$y_{start} = [f_0(\boldsymbol{\theta}_{start}), f_1(\boldsymbol{\theta}_{start}];$
$Y \leftarrow \{y_{start}\};$
```
/* main loop                                                                   */
```
**for** $t = 1, \ldots, T$ **do**
$\quad$ ```/* sample random element from the population                               */```
$\quad \boldsymbol{\theta}_t \sim \mathcal{U}(P_{t-1});$
$\quad$ ```/* mutate                                                                      */```
$\quad d \sim \mathcal{U}(0, |\boldsymbol{\theta}_t|);$ $\qquad\qquad\qquad\qquad$ ```// sample random dimension```
$\quad \hat{\boldsymbol{\theta}} \leftarrow copy(\boldsymbol{\theta}_t);$
$\quad \hat{\boldsymbol{\theta}}[d] \leftarrow \mathcal{U}(\Theta_d);$ $\qquad\qquad\qquad$ ```// sample a new value from the search space```
$\quad$ ```/* evaluate                                                                    */```
$\quad y_t = [f_0(\hat{\boldsymbol{\theta}}), f_1(\hat{\boldsymbol{\theta}})];$
$\quad Y \leftarrow Y \cup y_t$
$\quad$ ```/* update population                                                           */```
$\quad S(Y) = \{y' \in Y : \{y'' \in Y : y'' \succ y', y' \neq y''\} = \emptyset\};$ $\qquad$ ```// Pareto front```
$\quad P_t \leftarrow \{\boldsymbol{\theta} : y(\boldsymbol{\theta}) \in S(Y)\};$
**end**

**Algorithm 5:** Local Search

## E   Additional Results

In this section we present additional results from the experiments described in Section 4. Figure 10 shows the Pareto front of randomly chosen runs for each dataset for BERT-base, sorted by the training dataset size. RTE and MRCP are small datasets compared to other datasets, leading often to an unstable fine-tuning process.

Figure 11 and 12 shows the optimization trajectories for each methods using standard NAS and weight-sharing based NAS, respectively. Solid lines indicate the mean and shaded area the standard devication.

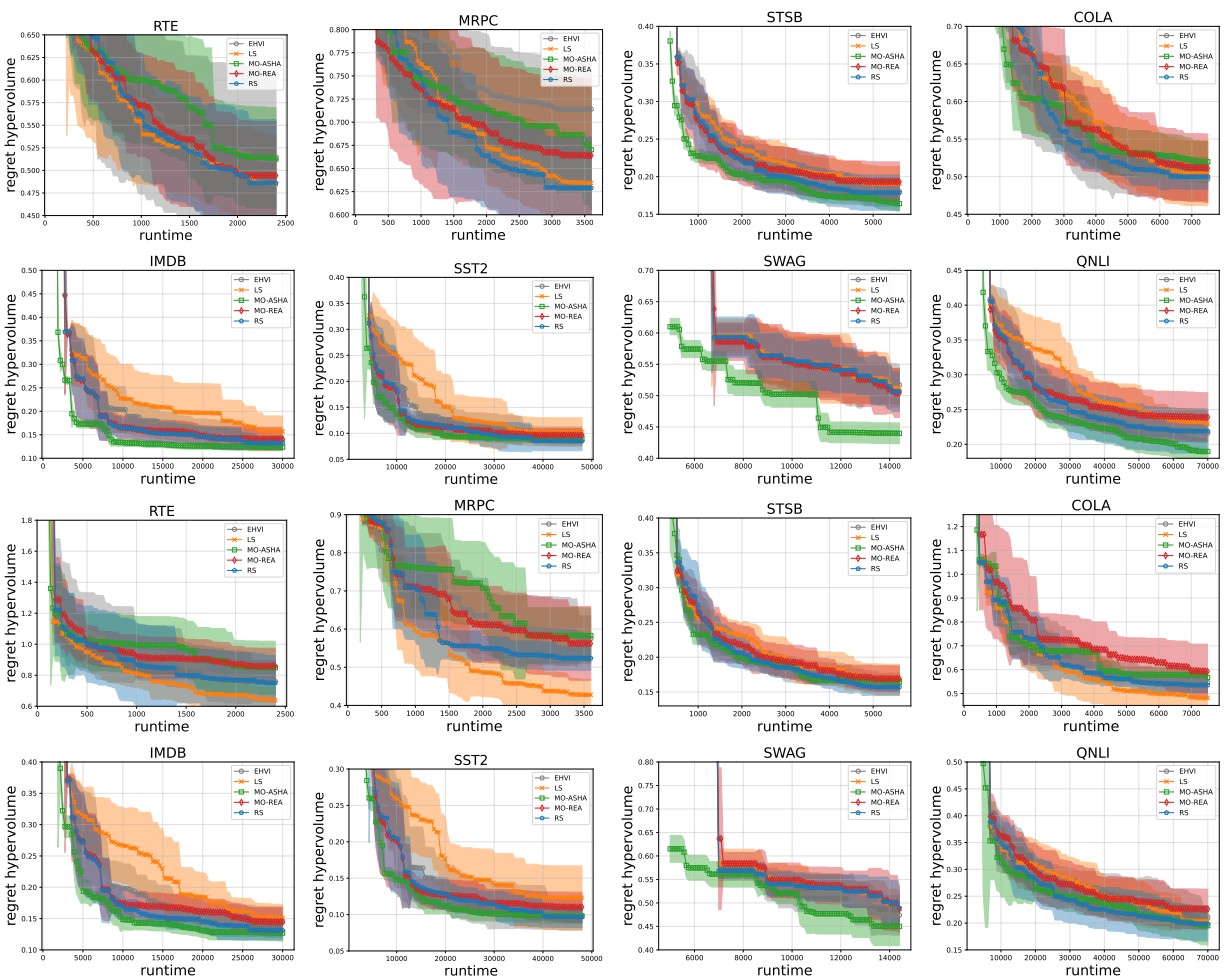

Figure 11: Hypervolume of standard NAS methods on BERT-base-cased (first two rows) and RoBERTa-base (last two rows).

## F    Quantization

Quantization (Dettmers et al., 2022; Dettmers & Zettlemoyer, 2023) is a powerful technique that significantly reduces the memory footprint of neural networks. However, its impact on latency is not immediate, especially when dealing with batch sizes that can not fit into the cache of the device Dettmers & Zettlemoyer (2023). With our flexible NAS framework we can simply replace objectives and directly optimize latency on the target device instead of parameter count.

Figure 13 left shows the Pareto set obtained with our NAS approach, where we optimize latency instead of parameter count on the COLA dataset across 3 different GPU types. Additionally, we evaluate the performance of the unpruned super-network with 8-bit (Dettmers et al., 2022) and 4-bit (Dettmers & Zettlemoyer, 2023) quantization. While quantization substantially reduces the memory footprint (Figure 13 right), it actually leads to worse latency. While quantization introduces a small overhead due to the additional rounding steps, the latency could potentially be reduced by optimizing the low-level CUDA implementation. Somewhat surprisingly using a int-8bit quantization leads to high performance drop on some hardware. NAS effectively reduces the sizes of weight matrices, leading to reduced GPU computation and, thus, is less hardware depend.

We can also apply quantization to sub-networks, making it orthogonal to our NAS methodology and offering further improvements to the memory footprint. Overall, these findings shed light on the trade-offs between

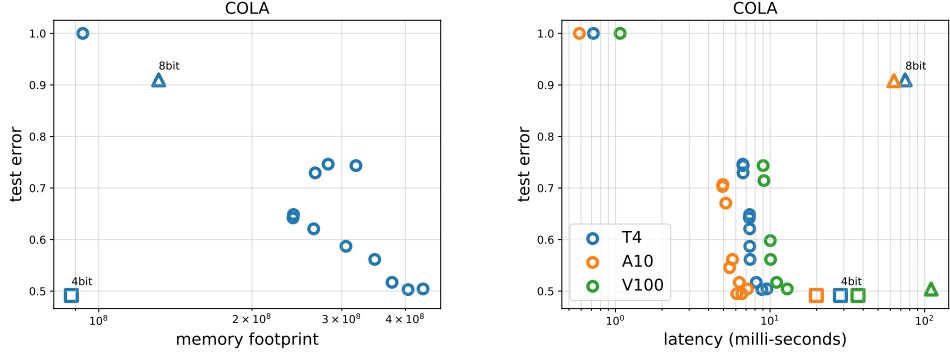

Figure 12: Hypervolume of search methods for weight-sharing based NAS on BERT-base-cased (first two rows) and RoBERTa-base (last two rows).

Figure 13: Test error versus memory footprint (left) and latency (right) on 3 different GPU types for the Pareto front found by our NAS strategy and the un-pruned network with 8bit and 4bit quantization.

memory footprint reduction and latency optimization. We leave it to future work to explore the connection between NAS and quantization.

