# OpenReview forum: "Structural Pruning of Pre-trained Language Models via Neural Architecture Search"
_TMLR — Accepted by TMLR_

### Review · Reviewer_NdN4 · 2024-05-23

**Summary Of Contributions:**

Though language models are very powerful in solving many natural language tasks, deploying them could be challenging because they require huge GPU memory and usually have high inference latency. This paper explores the model pruning to find a sub-part of the neural network, which is a trade-off between the resource consumption and the model capability.

The key idea of this paper is to use the Neural Architecture Search (NAS) technique to solve the model pruning problem. It first pre-trains the super network. Then, it generates many sub-networks by using different masks and fine-tunes those sub-networks. Finally, it selects those sub-networks that balance the model size and predict accuracy.

In summary, this paper makes three main contributions:

1. This paper proposes a new use case of NAS that can help solve the model pruning problem.
2. This paper proposes four search spaces of pruning language models that cover existing structural pruning strategies.
3. This paper proposes a new benchmark for multi-objective NAS. It evaluates different NAS algorithms on eight classification tasks.

**Audience:**

Yes

**Broader Impact Concerns:**

No ethical concerns.

**Claims And Evidence:**

Yes

**Requested Changes:**

1. The author should add some experiments or data to prove that the NAS-based solutions do have some unique advantages. For example, the NAS-based solutions have higher and acceptable model accuracy with the same pruning size, or they can do pruning much faster than traditional solutions.

2. The author should modify Sec 4.1 to define the hypervolume clearly.

**Strengths And Weaknesses:**

## Strengths:

1. The technology foundation of this paper is reasonable and sound. Model pruning and NAS share the same target: keeping only a part of the model while not harming the performance. Therefore, it is natural to apply the NAS technique to the model pruning problem.

2. The evaluation of this paper is very solid. This paper evaluates eight different datasets and two different language models with different pruning algorithms and settings. The substantial experiment result makes the conclusion quite convincing.

## Weaknesses:

1. This paper does not prove that the NAS-based solution has an obvious advantage over traditional solutions. According to the evaluation, traditional solutions have higher accuracy with small or medium pruning sizes. Though NAS solutions have higher accuracy with large pruning sizes, the accuracy loss could be 50% or more. To the reviewer's understanding, such accuracy loss is unacceptable.

2. The major metric Hypervolume is not well described. The description in Sec 4.1 is not clear, and it is hard to understand the meaning of the hypervolume and how we calculate it.

---

> ### Author Response · Authors · 2024-06-17
> **response to reviewer NdN4**
>
> We thank the reviewer for the valuable feedback and appreciate that they found the technological foundation of the paper to be sound and the empirical evaluation solid. We are also pleased that the reviewer found the conclusion of the paper convincing.
>
> 1. We’d like to highlight that the paper deliberately does not claim that neural architecture search has an obvious advantage over other state-of-the-art approaches. Instead, the contributions of the paper are twofold. First, we show that neural architecture search can be an efficient method for structural pruning, that, compared to other common pruning methods, has the mainly two advantages. First, it allows for a multi-objective approach to discover a Pareto optimal set of sub-networks, enabling users to select the optimal solution given their requirements. On the other hand, methods from the literature usually require defining a fixed constraint on model size, which is often hard to define given the non-linear relationship between model size and accuracy. Second, neural architecture search is a flexible framework that allows optimizing any metrics of interest to quantify the efficiency of a neural network, such as latency, energy consumption, or FLOPs.
>
> 2. We thank the reviewer for the suggestion and have added a more formal definition of the hypervolume to Section 4.1.

---

### Review · Reviewer_cEWC · 2024-06-03

**Summary Of Contributions:**

The paper considers pruning pre-trained language (PTL) models for downstream tasks requiring low latency and/or low memory footprint using various neural architecture search (NAS) techniques. As a byproduct, the paper stands as a benchmark for numerous NAS methods.
The paper starts with a comprehensive overview of the challenges raised by downstream applications of PTL and different NAS techniques in the first two sections. The third section lays down the NAS approach for structural pruning of PTL models using masks on multi-head attentions as well as fully connected layers. Several search spaces are presented with a quick overview of their relevance. A complete two-stage search procedure is detailed. Experiments delve into the relevance of NAS approaches for PTL models pruning. A first set of experiments demonstrate that a small search space is the most efficient in most benchmarks. Then a set of experiments study the performance of various NAS searches on this small search space. The authors present then comparisons of multistage NAS searches that also serve as an ablations study of the main approach presented earlier. Additional comparisons with quantization techniques are present in the Appendix.

**Audience:**

Yes

**Claims And Evidence:**

Yes

**Requested Changes:**

Questions:
- Why using the *first* units/heads? Since one can simply permute the columns of the next matrix multiplying the inputs, it seems that this choice is arbitrary. The argument of about the problematic concentration of the MEDIUM search space is great. But why not adding a permutation of the units/heads?
- An important point raised by the authors is the actual hardware constraints imposed on such models. I suppose one such constraint is compatible sharding. It would be great to expand on that since this can offer a principled approach to efficient pruning that satisfies the constraints (if I understand well the LARGE search space does not satisfy such constraints).

Suggested changes:
- Give a complete mathematical definition of the hypervolume as it is the main metric used in the benchmark.
- Maybe give more mathematical details on the NAS search that appears to perform the best in the main text and more details on all approaches in the Appendix to ensure that the paper is self-contained.
- Consider making paragraphs with take-aways from the experiments in bold, and then add additional details.
- Maybe consider making a streamlit link to ease the comparisons between approaches. A plot with more than three curves is generally hard to parse. (This is just a detail, feel free to ignore).
- Appendix F is quite interesting. It shows in particular the flexibility of the multi-objective approach to tackle issues beyond mere memory footprint. In fact latency is a leitmotif in the introduction that does not appear anywhere else. So I would suggest adding this section in th main text to strengthen the approach and maybe consolidate that experiment by adding for example Pareot frontiers and experiments on all benchmarks.

Details:
- On Fig. 1 MHA and FFN are acronyms which are defined much later, consider using the full names here.
- PTL (page 3) not defined.
- "For example, Sanh et al (2020) was trained..." (page 3) -> the author was not trained for 90 hours hopefully. I should read "the model of Sanh et al..."
- Maybe fully detail the Attention layer page 4. The paper has the potential to not only serve as a benchmark but also a gentle introduction to such pruning techniques, so it could be worth going all in on the the mathematical details.
- The notation U in the definition of $M_{neuron}$ is not given (it is way later but not at the time of the definition page 4).
- The notation $\circ$ in the application of the masks (page 4 second paragraph of Section 3.1) is not defined and also uncommon. A $\cdot$ may be more appropriate.
- The dimensions of the matrix multiplications in the definition of FFN (page 4) are ill-posed (you may use $X^\top$). Then when using $M_{neuron}$ it could be worth fulyl detailing mathematically what the $\circ$ means in this context (namely multiplications of rows/columns, depending on your choice of definition of FFN).
- Section 3.2.1 second sentence: "initialize" -> "initialized".
- Eq. 2 make parentheses larger for terms inside $\sigma$.
- Mention that $\mathcal{L}_{CE}$ is defined on some data (and maybe detail which one in practice).
- Last sentence of Section 3.2.2 not clear for non-knowledgeable readers.
- What do you mean by the dimensionality of the search space and its upper bound? (Section 3.3, SMALL search space second sentence).
- Put in italic/bold the NAS searches presented and used in Section 4.1.2.
- Add a suptitle on the first and third rows of e.g. Fig. 4 that says "BERT" or "RoBERTa" for ease of reading.
- In the last sentence of the paragraph Multi-Objective Search (page 9) there is a verb missing.
- In Appendix C, maybe give references to the different datasets used.

**Strengths And Weaknesses:**

Strengths:
- The paper is well written and serves as a gentle introduction to NAS approaches in a relevant context (pruning PTL models).
- The main multistage approach is well presented (additional questions presented in the request changes section).
- The set of experiments is quite thorough. Eight benchmarks are considered on two different PTL models. More than ten approaches are compared.
- The overall approach of the authors can serve as a benchmark for future research on NAS techniques. Benchmark papers can be extremely relevant in the field for further progress.
- Tackling pruning via a Pareto frontier can offer simple selection tools for end-users. This point could even be more highlighted. For example, the experiments in the Appendix on quantization show alternative multi-objective definitions that would be relevant for users.

Weaknesses:
- The exhaustive nature of the paper is remarkable. However, currently, take-away messages are not fully clear-cut. This is mostly a question of presentation. The current material is enough to raise the interest of peers.
- As the main contribution consists in a benchmark, the metrics should be fully defined in the paper rather than using some references.

---

> ### Author Response · Authors · 2024-06-17
> **response to reviewer cEWC**
>
> We thank the reviewer for the constructive and positive feedback, which helped us tremendously to improve our manuscript. We appreciate that the reviewer found our paper to be well-written and the experiments to be thorough. We also agree with the reviewer that benchmark papers such as this one are extremely helpful in making progress in the field.
>
> Following the reviewer’s suggestion, we added a detailed description of each neural architecture search method to the appendix to make the paper more self-contained. Also, as recommended by the reviewer, we added a more formal definition of the Hypervolume and visually highlighted the main takeaways in the conclusion paragraph of each experiment section. We hope these changes address the main weaknesses pointed out by the reviewer. Lastly, we fixed all typos and minor comments.
>
> Regarding the reviewer’s questions:
>
> 1. We use the first heads/units for the MEDIUM and SMALL search spaces to enforce a bijective mapping between architecture configurations and binary masks. For example, if we sample the configuration (layer=4, units=128, head=4) from the SMALL search space, we need to ensure that the resulting binary mask is deterministic to avoid introducing additional noise.
>
> 2. We agree that hardware constraints, such as compatible sharding, are an important point. Compared to other structural pruning approaches, multi-objective NAS represents a flexible framework that allows us to easily incorporate such constraints into the search process. Furthermore, we can directly optimize the metric of interest, such as latency, memory footprint, or energy consumption, without relying on additional gradient information.

---

> > ### Comment · Reviewer_cEWC · 2024-07-30
> > **Thanks for the answer**
> >
> > A few remaining comments:
> > - Symbol $\circ$ in 2nd paragraph of Sec. 3.1 is not defined.
> > - A figure illustrating the hypervolume would help. The definition is hard to grasp as it is. Since it is a key metric, it's worth taking time for the reader to grasp it.
> > - Make all ``conclusion'' paragraphs, real paragraphs (for example in 4.1.1, the conclusion is inside a paragraph).
> >
> > The authors have not expanded (i) the quantization part, (ii) the issue of hardware constraints. Such addendums are not necessary to accept publication, but beyond publication, such remarks could help the paper reach a broader audience.
> >
> > Thank you again.

---

> > > ### Author Response · Authors · 2024-08-06
> > > **addressed comments**
> > >
> > > Thank you for your comments, which we have addressed in the revised version of our paper. We agree that the topics of quantization and hardware constraints are intriguing. However, expanding on these subjects would significantly broaden the scope of the paper, making it challenging to include in the main text, and we consider a more detailed analysis as future work.

---

### Review · Reviewer_TYiE · 2024-06-12

**Summary Of Contributions:**

The presented paper considers the problem of structural pruning of the pre-trained language model from the neural architecture search perspective. The authors introduce four search spaces that define subnetworks. They state the corresponding bi-objective optimization problem, where generalization error and latency are used as evaluation criteria. The experiments consist of two parts: comparing different NAS methods and comparing the best NAS method with alternative structural pruning. The BERT-base and RoBERTa-base models are used for evaluation purposes and multiple text classification tasks. The experiments show that the NAS-based approach to structured pruning of the considered models is competitive or even better scalable than alternative approaches.

**Audience:**

Yes

**Broader Impact Concerns:**

No concerns about the ethical implications.

**Claims And Evidence:**

Yes

**Requested Changes:**

Major requests:
1) in the contribution list, the authors mention four search spaces but then map the existing structural pruning only to two of them. Please comment on why there are only two. This contribution point induces more intrigue than makes your results clear.
2) please add a caption to the two subplots with average ranks vs time steps
3) what are the reported ranks in Figure 5b? No definition in the paragraph "Multi-Objective Search"
4) the runtime comparison of the discussed methods is missing in the text. That is an essential factor in selecting the method for structural pruning. Please add such a comparison in section 4 and discuss the observed trade-off (if any) between the Pareto optimality of the obtained subnetworks and the computational complexity of the tested methods.
5) please add the source code link to reproduce your results


Minor requests:
1) "They demand significant GPU memory and exhibit high inference latency, making them impractical
for many real-world applications, for example when used in an end-point for a web service or deployed on an
embedded systems" - make consistent usage of article "an" and the following adjective+noun "embedded systems". The text has other typos; please proofread it carefully.
2) the notation $\circ$ in applying the binary mask is not defined
3) please make consistent y-label and text description for Figure 7 ("test performance" vs "test error"), or add explicit clarification that the smaller, the better
4) "Especially for higher pruning ratios, NAS outperforms ..." - what NAS? S-NAS or WS-NAS?

**Strengths And Weaknesses:**

Strengths:
1) relevant approach to the structural pruning procedure
2) extensive experiments that confirm the efficiency of the introduced approach
3) clear description of the problem statement and suggested algorithm

Weaknesses:
1) the main text and plots should be carefully proofread to sync them and use the same notation
2) only two not large-scale models are considered in experiments
3) absence of the runtime comparison for the considered approaches

---

> ### Author Response · Authors · 2024-06-17
> **response to reviewer TYiE**
>
> We appreciate the valuable feedback from the reviewer, which helped us improve our paper. We are pleased that the reviewer found the approach relevant and the experiments to be extensive and confirming the efficiency of our approach.
>
> To address the major requests from the reviewer, we have made the following additions and corrections to the paper. We also addressed all minor requests from the reviewer.
>
> 1. Thanks for pointing this out. We clarified that only these two search spaces resemble the kind of search spaces used by structural pruning methods.
> 2. We added a caption to Figure 5.
> 3. We clarified this in the Multi-Objective Search section (4.1.3).
> 4. All methods have access to the same amount of compute, given a fixed limit on wall-clock time. We clarified this in Section 4.2. Additionally, we added a paragraph discussing the runtime trade-off between S-NAS and WS-NAS in more detail.
> 5. All code is already available on GitHub but was omitted to preserve anonymity. We will add the link for the camera-ready version of the paper.

---

> > ### Comment · Reviewer_TYiE · 2024-06-20
> > **Response to the authors**
> >
> > Dear authors,
> >
> > Thank you for the revised manuscript! I confirm that my requests are addressed in the revised version. However, the equation for computing Hypervolume as "the M-th dimensional Lebesgue measure between the Pareto set Pf and the reference point r" could make the presentation clearer for a broad audience.

---

> > > ### Author Response · Authors · 2024-07-30
> > > **added missing equation**
> > >
> > > Thanks! We added the missing equation to define the Hypervolume

---

### Author Response · Authors · 2024-06-17
**rebuttal**

We thank all reviewer for the constructive and helpful feedback. We updated the current manuscript and respond to each reviewer in turn. We highlighted all changes in red for an easier comparison to the previous version.

---

### Decision · Action_Editor_4efv · 2024-08-02

**Recommendation:** Accept as is

**Comment:**

This paper explores neural architecture search (NAS) for structural pruning.
The goal of structural pruning is to identify sub-parts of a fine-tuned network that optimally trade off efficiency, such as model size and generalization performance. A key contribution of this paper is that it does not propose a pruning method with a fixed threshold. Instead, it advocates for a multi-objective approach that identifies a Pareto optimal set of sub-networks using the hypervolume metric. This allows for a flexible and automated compression process.

The reviewers commended the thorough experimental verification. Eight benchmarks are considered on two different pre-trained language models, and more than ten approaches are compared. The authors' overall approach can serve as a benchmark for future research on NAS techniques.

The paper is well-written and briefly introduces NAS approaches in the context of pruning pre-trained language models.

The reviewers find that the paper's exposition is extremely clear, the connection between NAS and pruning is well explained and explored, and the general approach (Pareto vs. focus on a single technique) could be of interest to end-users. Therefore, we propose this paper for a survey certification.

When preparing the final version of this manuscript, please consider the reviewers' suggestions for making this paper even more self-contained.

**Audience:**

Yes, model pruning is a topic that interests the TMLR audience. Additionally, the paper's discussion and findings on using neural architecture search (NAS) for structural pruning are of interest to the community.

**Claims And Evidence:**

The claims made in the submission are supported by clear evidence.